# Preattentive facilitation of target trajectories in a dragonfly visual neuron

Benjamin H. Lancer [1 ✉], Bernard J. E. Evans [1], Joseph M. Fabian[1], David C. O'Carroll[2] &
Steven D. Wiederman [1]

The ability to pursue targets in visually cluttered and distraction-rich environments is critical for predators such as dragonflies. Previously, we identified Centrifugal Small-Target Motion Detector 1 (CSTMD1), a dragonfly visual neuron likely involved in such target-tracking behaviour. CSTMD1 exhibits facilitated responses to targets moving along a continuous trajectory. Moreover, CSTMD1 competitively selects a single target out of a pair. Here, we conducted in vivo, intracellular recordings from CSTMD1 to examine the interplay between facilitation and selection, in response to the presentation of paired targets. We find that neuronal responses to both individual trajectories of simultaneous, paired targets are facilitated, rather than being constrained to the single, selected target. Additionally, switches in selection elicit suppression which is likely an important attribute underlying target pursuit. However, binocular experiments reveal these results are constrained to paired targets within the same visual hemifield, while selection of a target in one visual hemifield establishes ocular dominance that prevents facilitation or response to contralaterally presented targets. These results reveal that the dragonfly brain preattentively represents more than one target trajectory, to balance between attentional flexibility and resistance against distraction.

[1] School of Biomedicine, The University of Adelaide, Adelaide, Australia. [2] Department of Biology, Lund University, Lund, Sweden.
✉email: benjamin.lancer@adelaide.edu.au

Selective attention, the ability to respond to a selected subset of environmental stimuli, is important to many species across taxa and underlies a variety of behavioural tasks[1–6]. The study of target selection and attention has largely focused on vertebrates, but there is mounting evidence that insects are capable of attention-like computations[4,5]. Adult dragonflies are predatory pursuit specialists[7] that intercept prey mid-air with high success rates[8,9], by flying along interception trajectories based on predictive internal models[10,11]. We have identified a small-target, motion-sensitive, visual neuron in the dragonfly brain that exhibits selective attention via a winner-takes-all, competitive process when presented with paired targets, ignoring the unselected target as if it did not exist[12,13]. This neuron, termed Centrifugal Small Target Motion Detector 1[14] (CSTMD1), is then able to flexibly lock on to an attended target even when challenged by an abrupt-onset, high-contrast distractor[13] as well as dynamically switch attention between targets of equivalent or varying contrast[12,13].

CSTMD1 exhibits neuronal facilitation in response to a single target moving along a continuous path[15,16]. This facilitation manifests as a spotlight of gain enhancement that spreads predictively ahead of the target's current trajectory and is concomitant with suppression of surround locations in the receptive field[17]. This gain enhancement is thought to drive the neuronal response to saturation to render it less sensitive to transient changes in target saliency[18]. The interaction between attentional selection of a target and this facilitation mechanism for target trajectories is not yet known. One hypothesis is that facilitation subserves selection by boosting the signal of the attended target, resulting in a positive feedback loop similar to contrast-gain mechanisms observed in primate visual cortex[19–22]. Alternatively, facilitation may precede selection, with even representations of unselected targets becoming enhanced. Studies in humans and other primates reveal simultaneous tracking of multiple, independent targets, where each target generates its own spotlight of enhancement[23–25]. These spotlights can form at non-contiguous, independent spatial locations at the level of extrastriate occipital pathways and V1[23,25–27]. In the context of a dynamic scene where new opportunities and risks may become apparent over time, the ability to passively track multiple targets simultaneously may be desirable. However, it would remain critical to select only one target for the direction of action[28]. Otherwise, the animal could actuate an inaccurate average action vector[29–34].

Are both trajectories facilitated when one is selectively attended in the dragonfly target tracking system, or is it only the selected target's trajectory that is predictively facilitated? Here, we test this directly by recording CSTMD1 spiking activity in vivo in response to the simultaneous presentation of a pair of equally salient, rival targets. We assess the facilitation state ahead of the trajectory of the ignored target (i.e., the unattended target not represented in CSTMD1's spiking response). We show that when both targets are presented in the same visual hemifield, both are facilitated despite only one being selected. However, endogenous attentional switches between targets generate suppression on the trajectory of the previous selected target, similar to Inhibition of Return (IoR) observed in vertebrates[35]. In addition, we tested the extension of such mechanisms across the two sides of the insect brain when rival targets are presented in different visual hemispheres. We show that selection of a target in one hemisphere establishes ocular dominance and leads to long-lasting suppression of targets presented to the contralateral eye.

## Results

### Interaction between facilitation and target selection in the excitatory receptive field. To test for facilitation on unselected

trajectories, we presented Paired Primer Targets consisting of two 1.5° by 1.5° dark squares that moved upwards on the display at 50°/s, on rival trajectories (Fig. 1a, $T_1$ and $T_2$) within CSTMD1's excitatory receptive field. Primer targets were frequency-tagged by modulating contrast (at different frequencies) in order to elicit a frequency-locked response[13] from the selected target. This was used in subsequent analyses to determine the attended target. On any given trial, a Probe target was presented as a continuation of the trajectory of either $T_1$ or $T_2$ (but never both). We interleaved control trials consisting of either a single Local Primer (matched to the Probe trajectory) or Distant Primer (unmatched to the Probe trajectory), a Probe Alone (no primer), and Paired Primers (Fig. 1b). An extended descriptive pictogram of the visual stimuli and stimuli terminology is available in the methods section. To measure facilitation, we counted spikes within a 100 millisecond (ms) window, 50 ms offset from the Probe target onset to account for neural delays (Fig. 1b, dark green area). In the first set of experiments, we used short duration Primers ascending the receptive field at 50°/s for 400 ms (Fig. 1c). As previously observed[15,16], responses to a Probe that continued on the trajectory of a single Local Primer were significantly facilitated compared to the Probe Alone (Fig. 1c, red; Wilcoxon rank-sum test, $z = 11.84$, $p < 0.001$, $g = 2.36$ [1.99, 2.72]). At a 12° horizontal spacing between trajectory locations, we saw no significant effect on neuronal response to a Probe that appeared after a Distant Primer (Fig. 1c, blue; $z = 2.13$, $p = 0.08$, $g = 0.25$ [−0.03, 0.53]). Previously, surround suppression was observed at distances greater than 15° from the Primer trajectory[17]. However, here our targets were likely placed in between the locally facilitated and these more distant suppressed regions.

When Paired Primers were presented in CSTMD1's excitatory receptive field, the neuron responded to just one target of the pair[12,13]. However, the dragonfly cannot know in advance which of the Paired Primers would then be continuous with the follow-up Probe. Hence if only the selected Primer generates facilitation, we would expect the Probe to only be facilitated in a subset of the Paired Primer trials, leading to a broad distribution. This distribution would be equivalent to adding together those from the controls for single Distant and Local Primers. However, if the unselected Primer also generates neuronal facilitation, then the Probe should exhibit facilitation regardless of selection. We found that the Probe response following Paired Primers (Fig. 1c, purple) was significantly facilitated compared to both the Probe Alone ($z = 10.25$, $p < 0.001$, $g = 2.36$ [2.03, 2.70]) and Distant Primer ($z = 10.30$, $p < 0.001$, $g = 2.05$ [1.69, 2.41]) conditions, but not different from the Local Primer condition ($z = 0.47$, $p = 0.123$, $g = 0.19$ [−0.10, 0.49]). Frequency Polygons (Fig. 1d) show that the Paired Primer response distribution (purple line) more closely matched the Local Primer distribution (red line) than a theoretical equal combination of Local and Distant Primer responses (Merged Model, dashed purple). We repeated this experiment with longer duration primers (Long Primers) ascending the receptive field at 50°/s trajectory for 800 ms and observed similar results (Fig. 1e, f, with Paired Primer responses facilitated in comparison to Probe Alone ($z = 10.25$, $p < 0.001$, $g = 1.85$ [1.57, 2.13]) and Distant Primer ($z = 10.30$, $p < 0.001$, $g = 1.98$ [1.69, 2.28]), but not different from the Local Primer ($z = 0.47$, $p = 0.609$, $g = 0.02$ [−0.26, 0.22]). Thus, we observe a similar distribution of neuronal facilitation at the Probe location between single Local Primer and Paired Primer conditions. As the dragonfly cannot know which of the Paired Primers will be continuous with the Probe, both Primer targets must generate facilitation.

Do selected and non-selected targets generate the same magnitude of facilitation? To identify which Primer was selected on any given trial, we utilized frequency-tagging as previously

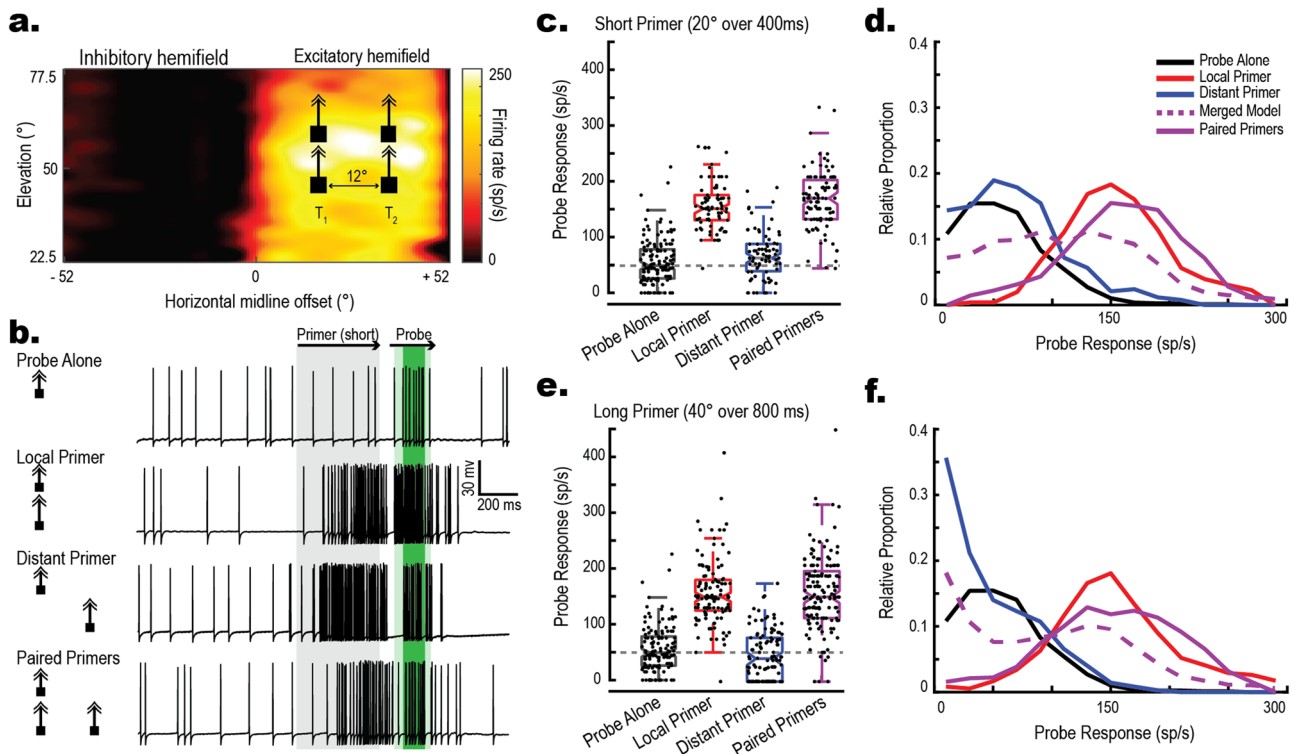

**Fig. 1 Paired targets generate facilitated responses. a** CSTMD1's Receptive Field with schematic stimulus pictogram superimposed (not to scale). The receptive field was mapped with a single 2°x 2° target moving horizontally (left-to-right) at 80°/s. The receptive field consists of two distinct zones, the excitatory hemifield (contralateral to the recording site in the axon) and the inhibitory hemifield (ipsilateral to the recording site.) **b** Left; Stimulus pictograms of four trial conditions ($T_1$ Probe locations shown; the same trials were also run using a $T_2$ Probe, i.e. mirrored). Top to bottom: Probe Alone, a 200 ms target Probe is presented alone. Local Primer, a Probe is spatiotemporally preceded by a facilitatory Primer on a matched trajectory. Distant Primer, the same Probe is preceded by a Primer on an unmatched trajectory (12° horizontal offset). Paired Primers, the same Probe is preceded by both a Local and Distant primer simultaneously. Right; Example spike trains drawn from the same neuron. Light green box indicates the 200 ms Probe period. Dark green box indicates the 100 ms analysis window. **c, d** Results for the presentation of Short Primers lasting 400 ms. **c** Box-and-whisker plots with overlaid swarm plots showing Probe response to the varying conditions. Each dot represents an individual trial (389 total trials across 15 dragonflies). Spike rate was calculated from a 100 ms period 50 ms following the onset of the probe (**b**, dark green window). **d** Frequency polygons of the same data. Merged Model (purple, dashed) represents the combined Local (red) and Distant (blue) Primer distributions. The empirical Paired Primer distribution (purple, solid) more closely matches Local Primer condition than a combination of local and distant primers, indicating overall facilitation. **e, f** The experiment with Long Primers (800 ms) exhibits similar results (507 total trials across 15 dragonflies), presented as in **c, d**. Note Probe Alone data is repeated from **c**.

validated in CSTMD1[13]. As illustrated in Fig. 2a–c, each Primer target's contrast was modulated (Weber = 0.22 to 1) at a unique frequency (11, 15 Hz, square waveform), resulting in frequency-locked neuronal responses. This allowed us to identify the selected Primer in ~70% of Paired Primer trials using a Selectivity Index[13] (detailed description in *Methods*). As previously observed[13], in approximately 30% of trials frequency-tagging did not elicit sufficient modulation for us to confidently identify the selected target, so these trials were excluded from further analysis. Figure 2d shows the assignment for each Paired Primer trial. If the selected trajectory was matched to the Probe, then it was categorized as Local Selection (gold, 126 trials), or as Distant Selection if unmatched (cyan, 128 trials). We found no statistically significant difference in CSTMD1's response to the Probe between Local Selection and Distant Selection trials (Fig. 2b, gold vs cyan; $z = 1.92$, $p = 0.052$, $g = 0.25$ [0.006, 0.501]), supporting the earlier conclusion that similar facilitation was generated along both target trajectories, irrespective of target selection.

**Do attentional cues influence generation of facilitation?** We recently reported that presenting a spatiotemporally preceding Cue on a single trajectory before paired targets biased selection towards that target[13]. Does the addition of such a Cue affect the

generation of facilitation by Paired Primers? Fig. 3a, b show data where one Primer preceded the appearance of the second. This Cue was always matched to the subsequent Probe (either $T_1$ Cue and $T_1$ Probe; or $T_2$ Cue and $T_2$ Probe). In most trials (~80%) the Cue induced Local Selection (Fig. 3a, gold), and the Probe then exhibited strongly facilitated responses compared to Probe Alone ($z = 10.32$, $p < 0.001$, $g = 2.12$ [1.79, 2.45]), quantitatively similar to the facilitation induced by the single Local Primer condition ($z = 6.48$, $p = 0.433$, $g = 0.009$ [−0.26 0.28]). However, we also saw Distant Selection in a smaller number of trials (25 trials, ~20%) presumably reflecting a switch in attention away from the Cue to the more novel target (Fig. 3a, cyan). Here, we then observed a reduction in the Probe response compared to both Local Primer alone ($z = 1.93$, $p = 0.009$, $g = 0.39$ [−0.05, 0.82]) and Local Selection ($z = 2.11$, $p = 0.004$, $g = 0.40$ [−0.04, 0.85]). Nevertheless, while weaker than single local primers, the response was still much stronger compared with the unfacilitated Probe Alone ($z = 4.78$, $p < 0.001$, $g = 1.64$ [1.18, 2.11]), indicating strong facilitation in at least a subset of trials.

What could account for weaker facilitation on the cued trajectory when the non-cued, distant Primer was selected? In primate neurophysiology and human psychophysics, switching attention from one location to another is associated with a suppressive signal known as Inhibition of Return[35] (IOR). Such

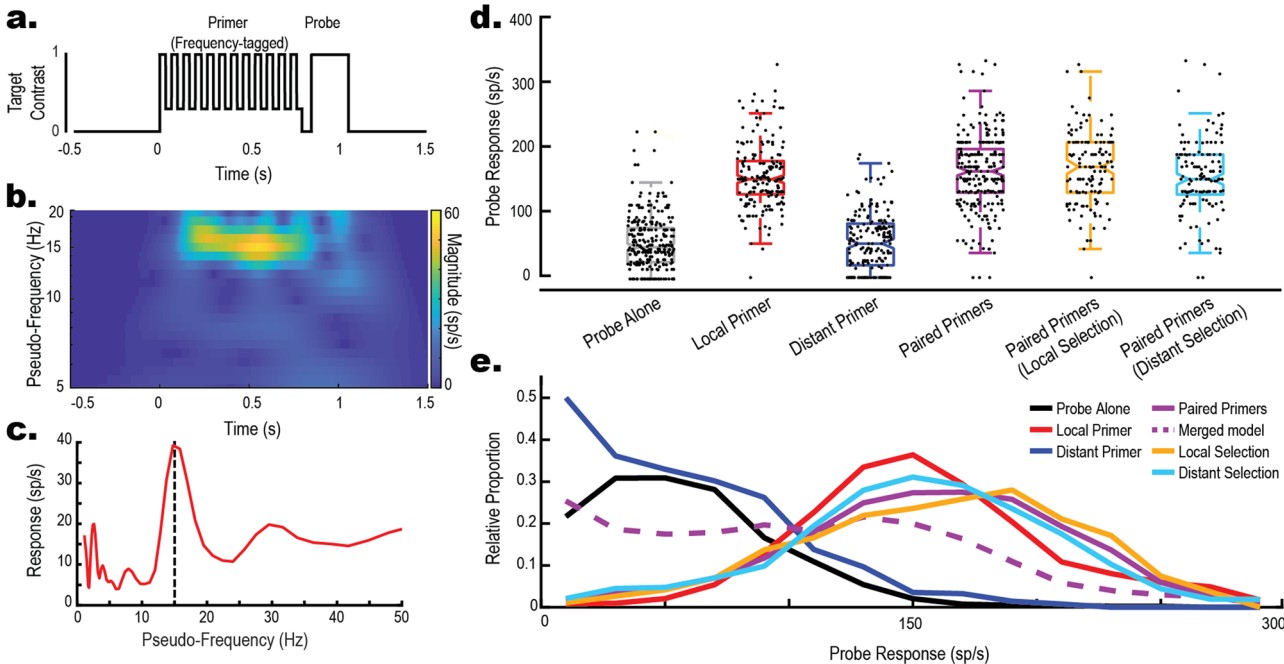

**Fig. 2 Facilitation is generated on unselected trajectories.** We applied frequency-tagging to Primers to determine which was selected on any individual trial. **a** An example of frequency-tagging with the modulation of one Primer contrast at 15 Hz before a Probe. **b** Wavelet scalogram of the spike activity (Inverse Interspike Interval) in response to a single example trial reveals a frequency-locked response at 15 Hz. **c** Time-collapsed wavelet scalogram (**b**) reveals a peak around 15 Hz. **d** Box-and-whisker with overlaid swarm plots illustrating Probe response, for Long (800 ms) and Short (400 ms) Primers combined (896 total trials across 15 dragonflies). For comparison, Probe Alone, Local, Distant and Paired Primers are shown (i.e. same data as in Fig. 1). Local Selection and Distant Selection box plots are the Paired Primers (purple) categorised by the Selectivity Index. We observe similar distributions between Local and Distant selection trials. **e** Frequency Polygons illustrating the distributions of each condition. Local Selection (gold) and Distant Selection (cyan) more closely match Local Primer (red) than the Merged Model (dashed purple) confirming that the Probe target is facilitated with either Local or Distant Selection.

inhibition prevents the attentional system from returning to previously assessed locations, allowing efficient visual search[36,37]. In our data, selection returns to the originally cued trajectory as the selected Primer disappears and the single Probe appears. Therefore, we propose that although both targets of a presented pair generate facilitation, an attentional switch generates some inhibition to the initially selected and facilitated trajectory, in a manner similar to Inhibition of Return. However, in contrast to previously described IOR, we observe the return response is a weaker facilitation, i.e. still enhanced compared to the Probe alone.

Figure 3c, d show data when the Cue is unmatched to the Probe (either $T_1$ Cue and $T_2$ Probe; or $T_2$ Cue and $T_1$ Probe). While we observed overall facilitation of the probe response (i.e., compared to Probe Alone) for both Local Selection ($z = 2.75$, $p < 0.001$, $g = 1.42$ [0.90, 1.95]) and Distant Selection ($z = 1.74$, $p < 0.001$, $g = 0.70$ [0.43, 0.98]) In the majority of trials (93 trials, ~83%; Fig. 3b cyan) where the Cued, Distant Primer was selected, Probe responses were broadly distributed and elicited reduced overall facilitation compared to a simple local primer ($z = 6.12$, $p < 0.001$, $g = 1.14$ [0.81, 1.47]). Intriguingly, some trials in this condition exhibited facilitation and others did not, matching the merged model ($z = 0.17$, $p = 0.632$, $g = 0.03$ [−0.21, 0.28]) combination of single Local and Distant Primers (Fig. 3b frequency polygons, dashed purple). Even in trials where the Local Primer was selected from the Pair despite the distant Cue (18 trials, ~16%; Fig. 3b gold), the distribution matched the merged model ($z = 1.28$, $p = 0.244$, $g = 0.44$ [−0.04, 0.93]) but with a small bump of facilitation at the right tail. It is interesting that the Distant Cue condition is the only stimulus condition to

exhibit such a broad distribution resembling the merged model. This reveals that facilitation generated by the introduction of a novel (i.e., not cued) target appearing during tracking of a cued target is a binary effect. That is, some trials matched the facilitation generated by a Local Primer, whilst other trials matched the Distant Primer response, resulting in a broad distribution. A feasible but speculative explanation for this is that the attentional system may actively suppress *both* the response and the facilitation to targets appearing during ongoing tracking (i.e., when the system is already attending), in contrast to the facilitation of both the selected and unselected targets when the targets appear together. Such an attentional suppression mechanism, sensitive to stimulus history, would result in stochastic trial-by-trial results due to natural variability in CSTMD1's response onset to a target and previously observed response delays when presented with rival target pairs[12]. This could lead to a binary outcome where a weak response occurs in those trials where the novel target is supressed following prior establishment of attention by the cue and suppression mechanisms are engaged (i.e. the response has locked on[13]), and stronger responses if the cue fails to establish strong attention before the rival target appears, such that pre-attentional facilitation occurs at both possible locations, as in un-cued trials (Fig. 2). Prior results have shown that the effect of a cue is an overall bias towards the cued target (over many trials) rather than a trial-by-trial guarantee[13], which would be explained by stochasticity in the time it takes for attention to lock on and engage suppressive mechanisms. However, the biasing effect of a cue diminishes over 1000 ms[13] suggesting that this suppression effect acts as a hurdle to novel, transient distraction rather than complete suppression.

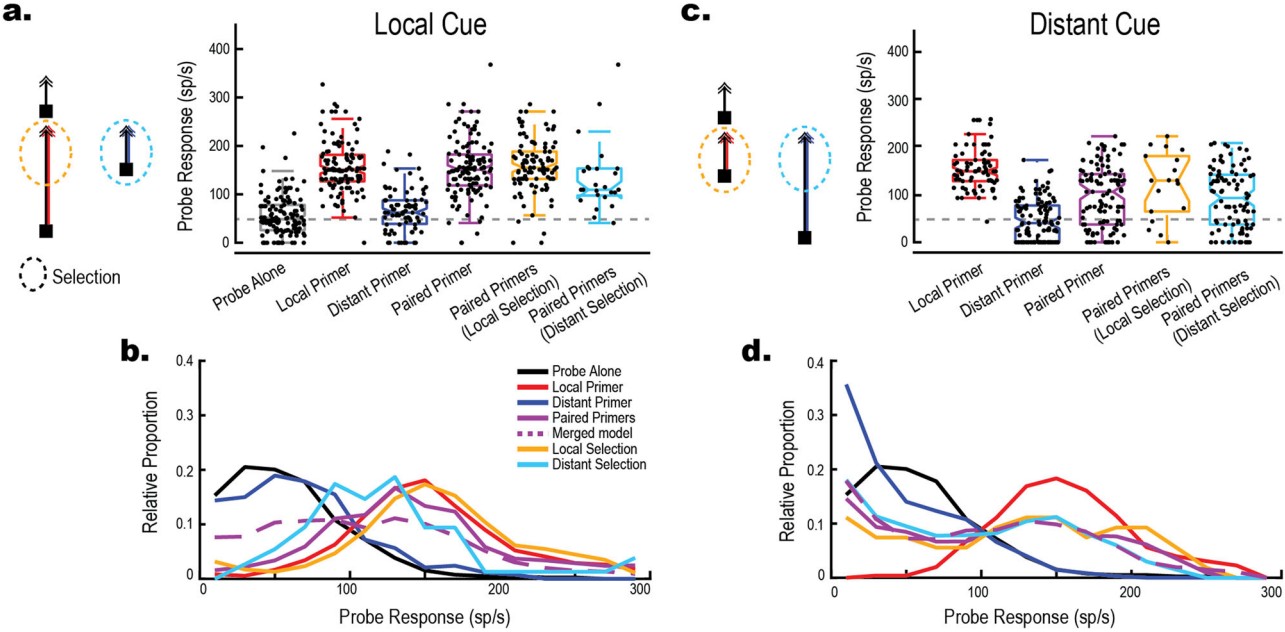

**Fig. 3 A Cue for selection can modulate facilitation. a** Stimulus pictograms and boxplots illustrating the neuronal response to the probe for trials where the cue was matched to the Probe location (Local). Total 122 Matched trials across 15 dragonflies, control data as from previous figures reproduced for comparison. Grey dashed line indicates the Probe Alone mean. In trials where the Cue is successful (Local Selection, gold) we observe facilitation. However, in trials where the Cue is ignored (Distant Selection, cyan) we observe suppression of the local path. **b** Frequency polygons showing Probe response distributions for the conditions in A. **c** Trials where the Cue was unmatched to the Probe location (Distant). Total 111 unmatched trials across 15 dragonflies, control data as from previous figures reproduced for comparison. Grey dashed line indicates the Probe Alone mean (Distribution in A). Unmatched trials reveal Paired Primer responses similar to the broad distribution of the Merged Model (Dashed purple line), regardless of whether Paired Primers were categorized as Local Selection (gold) or Distant Selection (cyan). **d** Frequency polygons showing Probe response distribution for the conditions in **c**.

**Target selection and facilitation in the binocular receptive field**. CSTMD1's receptive field contains two discrete hemifields[14], one excitatory and one inhibitory (Fig. 1a). Our previous work found that when two targets are presented simultaneously in each visual hemifield, CSTMD1 responses were strongly suppressed on average[38]. However, this study was undertaken before the realization that CSTMD1 showed selective attention in individual trials for paired targets presented in the excitatory hemifield[12]. We therefore again presented simultaneous target pairs consisting of one target in each of the inhibitory ($T_i$) and excitatory ($T_e$) hemifields, with individual examples shown in Fig. 4a (the analysis window shaded in green). Targets moved up the display monitor at 25°/s for 1 s. A subset of these trials included an additional 0.5 s Cue target to bias selection, as described earlier[13].

Figure 4b shows inhibitory or excitatory responses to single targets (either Short or Long trajectories), dependent on the corresponding hemifield. In response to Paired targets without a Cue, ~80% of trials elicited suppression of spiking activity (< 25 sp/s), showing a preference for selecting the target in the inhibitory hemifield. This aggregate data reveals significant differences between the Paired Target conditions. Cueing for $T_e$ elicited stronger responses than the uncued case ($z = 8.87$, $p = 0.008$, $g = 2.08$ [1.12 3.16]), revealing more frequent selection of the excitatory target if it precedes the other. Cueing $T_i$ elicited weaker responses compared to the uncued case ($z = 6.46$, $p = 0.001$, $d = 1.11$ [0.27 2.01]), with more frequent selection of the inhibitory target. In each individual neuron, we saw both inhibitory and excitatory responses to Paired Targets in individual trials with the corresponding Cue (Fig. 4c $T_i$ Cue: blue dots, $T_e$ Cue: red dots). These data show that selection between Paired targets presented either side of the visual midline can be biased by a preceding target trajectory.

A mechanism that might underlie the biasing of a preceding Cue target is the generation of spatial facilitation at earlier levels of processing, i.e., prior to the synaptic sign inversion that gives rise to inhibition in one hemifield. Previous experiments on facilitation have largely been confined to CSTMD1's excitatory receptive field[15,16,18]. However, in one experiment we observed that an inhibitory primer heading towards the excitatory hemifield still elicited facilitation across the visual midline[17]. This revealed that facilitation information could transfer across brain hemispheres. Thus far, no studies have examined facilitation for trajectories constrained within the inhibitory hemifield.

Do targets in the inhibitory hemifield generate neuronal facilitation similar to excitatory targets? We presented CSTMD1 with a Probe in the inhibitory region of the receptive field, either alone or following either a Short (400 ms) or Long (800 ms) Primer (Fig. 4d) ascending trajectories that remain confined to one hemifield. CSTMD1 exhibits a spontaneous spike rate of 5–10 sp/s, which is reliably driven to 0 sp/s with the presentation of a small moving target within the inhibitory receptive field (Fig. 1a)[10]. If inhibitory trajectories are also facilitated, we would expect facilitation of inhibition where spiking activity to a Probe continuing after an inhibitory Primer are driven to 0 faster than the inhibitory responses to that Probe presented alone. We found that a brief (200 ms) Probe target did not generate significant inhibition relative to the spontaneous spike rate ($p = 0.956$). However, the same Probe elicited inhibitory responses when preceded by either a Short 400 ms ($z = 4.12$, $p < 0.001$, $g = 1.10$ [−1.60, −0.70]) or Long 800 ms ($z = 4.91$, $p < 0.001$, $g = 1.13$ [−1.65, −0.64]) Primer, on the same trajectory (Fig. 4f). This decrease in spike rate is likely the result of an increased inhibitory drive from presynaptic excitatory facilitation, thus we refer to it as a facilitation of inhibition. In line with several earlier studies

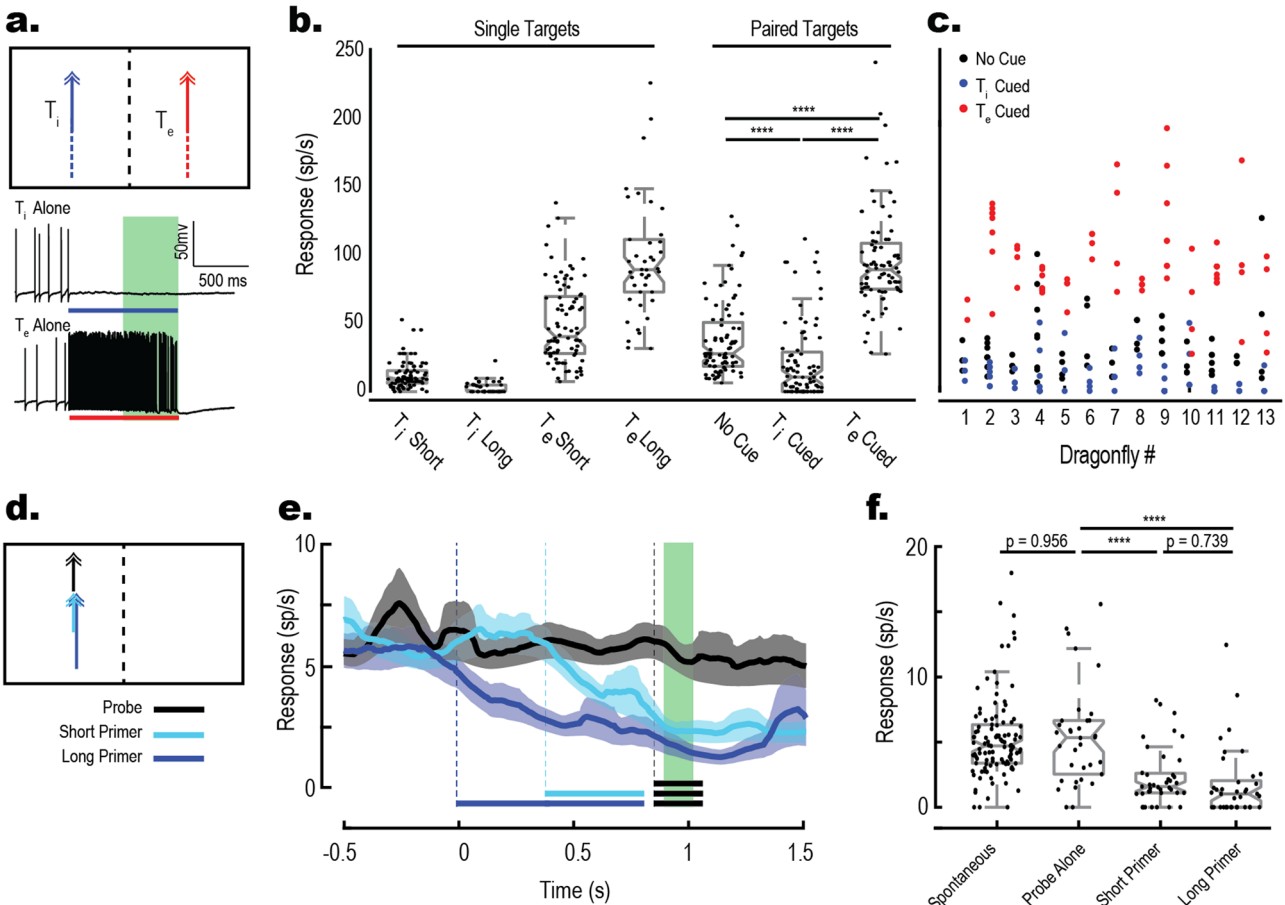

**Fig. 4 Target selection and facilitation in the inhibitory receptive field. a** Targets vertically ascend a display within CSTMD1's excitatory or inhibitory receptive field, either individually or as a simultaneous pair. Targets are separated by 50° to avoid the 10° wide region of binocular overlap. Target trajectories can be Short, Long or used as Cues, illustrated with dashed lines. Example traces illustrate CSTMD1 responses to $T_i$ and $T_e$. $T_i$ generates inhibition, $T_e$ generates excitation. **b** Boxplots with overlaid swarm plots illustrating the mean neuronal response over a 500 ms window (green shaded region in A). Total 480 trials across 13 dragonflies. In Paired Target conditions, neuronal responses are broadly distributed, with some trials exhibiting inhibition and others exhibiting excitation ($p < 0.001$). Compared to the Paired condition, a preceding $T_i$ Cue shifts responses towards inhibition ($p = 0.001$) and a $T_e$ Cue towards excitation ($p = 0.008$). **c** Responses in each of the 13 neurons to simultaneous paired target trials (black), $T_i$ Cued trials (blue) and $T_e$ Cued trials (red). The majority of neurons show inhibitory responses to Paired Target trials without cueing, however each neuron is able to respond to either the excitatory or inhibitory target with the appropriate Cue. **d** Stimulus pictograms illustrating three conditions; Probe Alone (Black line only), Probe preceded by a short Primer (Black and Cyan line), and Probe preceded by a long Primer (Black and Blue). Trajectories are illustrative as they overlie in experiments. **e** Averaged response across all trials (shaded region is standard error, 110 trials across 11 dragonflies) reveals the time course of inhibition. Presentation of a target within the inhibitory receptive field drives neural activity below spontaneous rates (−0.5 to 0 s). **f** Boxplots with overlaid swarm plots show that Probe response is reduced when paired with either a Short (400 ms) or Long (800 ms) primer. Spontaneous spike rate is also illustrated and was measured over an equivalent time period before the beginning of each trial. Statistical testing via Wilcoxon rank-sum test.

suggesting that facilitation is largely complete within 400 ms of stimulus onset[15–18], we found no difference in the amount of increased inhibition elicited by the Short or Long Primer conditions (Fig. 4e, f, $z = 0.39$, $p = 0.739$).

To examine how facilitation and selection interact across hemispheres, we then presented both paired or cued paired Primers on both inhibitory ($T_i$) and excitatory ($T_e$) target trajectories, analogous to the experiments presented in Figs. 1 and 3. We observed facilitation (compared to probe alone) of an excitatory-hemifield Probe following the selection of a matching primer from Paired Primer presentations (Fig. 5 a, gold conditions; Short Primers: $z = 4.22$ $p < 0.001$, $g = 1.88$ [1.23, 2.57]; Long Primers: $z = 3.95$, $p < 0.001$, $g = 1.70$ [1.08, 2.36]), irrespective of the Cue location in Cued trials (Matched: $z = 4.06$, $p < 0.001$, $g = 1.23$ [0.75, 1.72]; Unmatched: $z = 5.11$, $p < 0.001$, $g = 1.57$ [1.07, 2.10]). These results show that when selected, an

excitatory hemifield Primer can generate facilitation despite the existence of a simultaneous inhibitory-hemifield stimulus. However, when the inhibitory-hemifield Primer was selected from a pair (Fig. 5a, cyan conditions), we observed a sharp suppression of the subsequent $T_e$ Probe response (Short Primers $z = 2.77$, $p = 0.011$, $g = −0.82$ [−1.61, −0.05]; Long Primers $z = 2.59$. $p = 0.011$, $g = 0.09$ [−0.66, 0.47]). Intriguingly, we also observed suppression of $T_e$ Probe responses following $T_i$ Primer alone (Fig. 5a, Blue; Short Primers $z = 5.12$, $p = 0.001$, $d = 1.07$ [−1.54, −0.61]; Long primers $z = 5.01$, $p < 0.001$, $d = −1.12$ [−1.60, −0.65]) although primer and probe were never simultaneously presented, indicating that this suppression is not the result of a selection process. Instead, these data show that responses to an inhibitory target suppress CSTMD1's ability to respond to a subsequently presented excitatory Primer, even after a 50 ms pause.

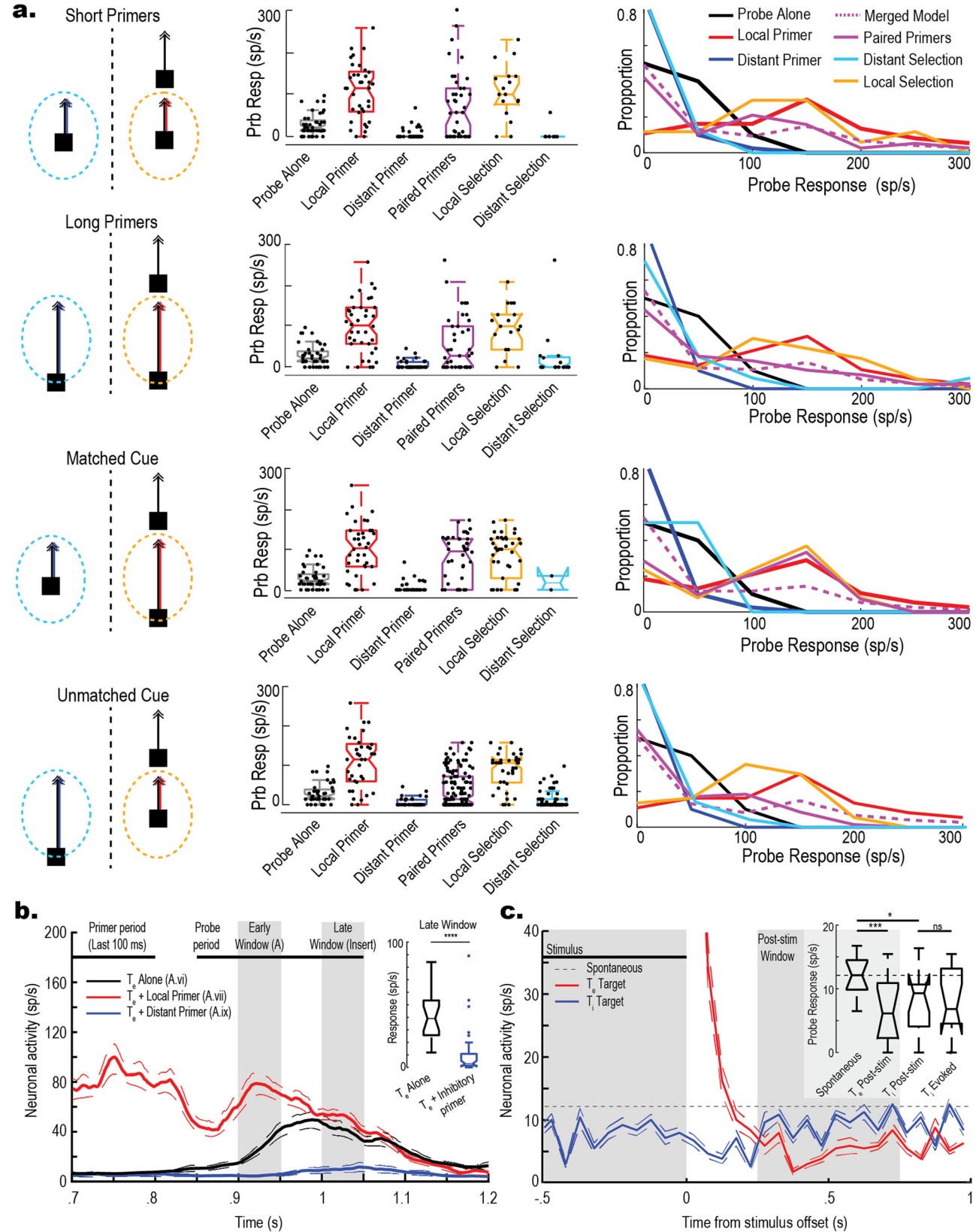

**Long-lasting cross-hemispheric inhibition establishes ocular dominance.** To examine this long-lasting inhibition further, we analysed conditions in which the excitatory Probe interacted with either excitatory or inhibitory single Primers (Fig. 5b). We observed that $T_e$ Probes elicit early facilitated responses when preceded by Local Primer (Fig. 5b red line), compared to the

Probe Alone (black Line). The $T_e$ Probe Alone already begins to facilitate within 100 ms once its path commences, as expected from prior findings[18]. When the $T_e$ Probe is preceded by a Distant ($T_i$) Primer it is significantly inhibited compared to the $T_e$ Probe Alone in both an early window (*50 ms post Probe onset, as noted above*) and a late (150 ms post Probe onset) analysis

**Fig. 5 Target selection establishes ocular dominance. a** Stimulus pictograms illustrate relevant trials. Boxplots illustrate the neuronal response to the probe. Frequency Polygons illustrate the Probe Response distributions. The Local Selection condition (gold) is consistently enhanced compared to Probe Alone (black) indicating facilitation, however both Distant Primer (blue) and Distant Selection (cyan) conditions are inhibited. Total 351 trials across 12 neurons. **b** Average inverse ISI plots reveal the time course of the neuronal response to an excitatory ($T_e$) Probe. Total 195 trials across 12 neurons. $T_e$ Probes elicit early facilitated responses when preceded by a $T_e$ Primer (red line) compared to when alone (black line). However, even the $T_e$ Probe alone is able to reach a facilitated state after approximately 100 ms. When preceded by a $T_i$ Primer the $T_e$ Probe is significantly inhibited (blue line) as in **a**, but this inhibition remains even in a late analysis window 150 ms after the onset of the Probe (Insert, $p > 0.001$, Wilcoxon rank-sum test), well within the time necessary to self-generate facilitation independently of prior priming. This observed suppression indicates cross-hemispheric inhibition evoked by a Primer in the inhibitory visual field lasts for at least 200 ms, although some individual trials are able to break through this suppression (insert, blue outliers).
**c** Averaged PTSH reveals post-stimulus inhibition occurs for both excitatory and inhibitory stimuli. Data reanalysed from Fig. 4 (Fig. 4c, second and fourth from top); 160 total trials across 12 dragonflies. Following the stimulus offset (time = 0) of either an excitatory (red line) or inhibitory (blue line) small moving target, CSTMD1's spike rate is supressed for an extended period (at least 1 s) in comparison to the spontaneous firing rate (black dotted line). Insert: Boxplots illustrating the average per-cell response across conditions. Spontaneous vs. $T_e$ Post-stim $p < 0.001$; Spontaneous vs. $T_i$ post-stim $p = 0.019$; Evoked vs. $T_i$ post-stim $p = 0.917$. Paired-sample t-tests were based on average neuronal responses (12 neurons).

window ($z = 5.04$, $p < 0.001$, $g = 1.77$ [1.15, 2.42]). This is despite the late analysis window being set sufficiently after the time required for the $T_e$ probe in unprimed trials (black line) to reach a similar degree of facilitation to the fully primed condition induced by the Local Primer. Thus, we observe profound long-lasting inhibition evoked by a Primer in the inhibitory hemifield (lasting at least 200 ms), masking our ability to examine facilitation of a subsequent $T_e$ Probe. It is not clear whether following an Inhibitory Primer, there is no facilitation at the Excitatory Probe, or whether local facilitation still exists but is masked by this long-lasting inhibition. In either case, target trajectories in the opposite hemifield to the selected stimulus do not generate net facilitation in CSTMD1's response.

To further examine post-stimulus excitation or post-stimulus inhibition, we re-analysed the single-target trials presented in Fig. 4b, focusing on a time window after the stimulus offset (500 ms window starting 250 ms after target disappearance). CSTMD1's responses exhibited significant post-excitatory inhibition following the offset of a $T_e$ target (Fig. 5c; $T_e$ post-stimulus window vs Spontaneous, $z = 2.57$, $p < 0.001$, $g = 0.54$ [−0.27, 1.37]). Such post-excitatory inhibition is a common property of spiking neurons, typically associated with the build-up of slow potassium currents during excitation resulting in sustained hyperpolarization. What occurs following a target placed within the inhibitory hemifield? Many neurons exhibit a similar post-inhibitory rebound of enhanced sensitivity or spontaneous firing following an inhibitory signal, contributing to a Motion After Effect in visual circuits[39]. However, in CSTMD1 we observed long-lasting inhibition following a $T_i$ target (Fig. 5c; $T_i$ post-stimulus window vs Spontaneous, $z = 2.11$, $p = 0.019$, $g = 1.52$ [0.64, 2.49]), which matched the suppression generated by the presence of a target ($T_i$ stimulus window (−0.5 to 0 s) vs post-stimulus window (0.25 to 0.75 s), $z = 0.14$, p = 0.917). We did not observe increased sensitivity following the disappearance of an inhibitory stimulus. As noted above, we observed the opposite, where a normally excitatory target was unable to generate a response despite appearing 50 ms following the offset of an inhibitory target (Fig. 5a, Distant Primer Condition). In contrast, an excitatory target is able to re-engage spike generating mechanisms if it appears during the post-excitatory rebound of another excitatory target (Fig. 5a, Local Primer condition), although it remains moderately inhibited if the second target does not appear within the facilitated region of the first[15]. Thus, CSTMD1 responds to stimulus offset with robust inhibition, regardless of the valence of the stimulus.

## Discussion
We have shown that when a pair of targets occupy the excitatory receptive field, each target generates its own spotlight of spatial facilitation, thus encoding the trajectory of both. One of these is

selected for representation in CSTMD1's spiking response, suggesting that computations underlying target facilitation precede competitive selection in the dragonfly attentional network. We refer to this as preattentive facilitation to emphasise the distinction against gain-increase models of attention which similarly facilitate neuronal responses, however, to only attended features[19,21,22,40]. As facilitation has been observed in other small target motion detectors (STMDs) within the lobula complex (presumed to be upstream of CSTMD1)[17], it is likely that this is a property of this earlier STMD pathway. These results resemble primate studies showing simultaneous tracking of multiple independent targets[41], where each target generates an independent spotlight of enhancement[23–25]. However, in contrast to our findings, a primate's ability to track multiple targets is independently divided between left and right hemispheres[42]. For humans, it is easier to track one target presented to each eye than it is to track a pair of targets in the same eye[43] and attentional modulation is attenuated for multiple targets presented in the same hemifield[43,44].

Our earlier work proposed that facilitation enhances the encoding of a target in variable visual conditions by increasing contrast sensitivity, inducing directional selectivity, and driving the neuron to saturation[17,18], allowing CSTMD1 to be both feature invariant and highly tuned depending on the situation. Thus, once a target is facilitated, it is able to generate a robust neuronal signal even during a high-speed pursuit where the target's local contrast against the immediate background may be highly variable due to background clutter. Additionally, the predictive encoding of a target's future position[17] may be involved in predictive behaviours performed by dragonflies before and during pursuit[9–11,45]. However, we have shown that facilitation is generated even for nonselected and inhibitory targets. Why should such distractors be similarly enhanced? We suggest that facilitation may also act as a gatekeeping mechanism for selective attention (Fig. 6). By initially suppressing targets in the surround, Predictive Gain Modulation ensures attention is not recruited by abruptly appearing, transiently highly-salient stimuli[13]. However, if a stimulus remains on a consistent trajectory generating its own facilitatory spotlight, it can become a viable target for stimulus selection. Therefore, the various components of Predictive Gain Modulation (spatial facilitation, induced directional selectivity, suppressive surround) work together to gatekeep attention by blocking distracting, transient and inconsistent stimuli, whilst enhancing stimuli on consistent trajectories, likely to represent potential prey, predators, or conspecifics.

This generation of facilitation on unselected paths can account for the diminishing effect of cueing previously observed[13]. A cued target benefits from facilitation generated by the cue, while a non-cued target appears within the supressed surround, thus biasing selection to targets continuing on the cue trajectory. Previously,

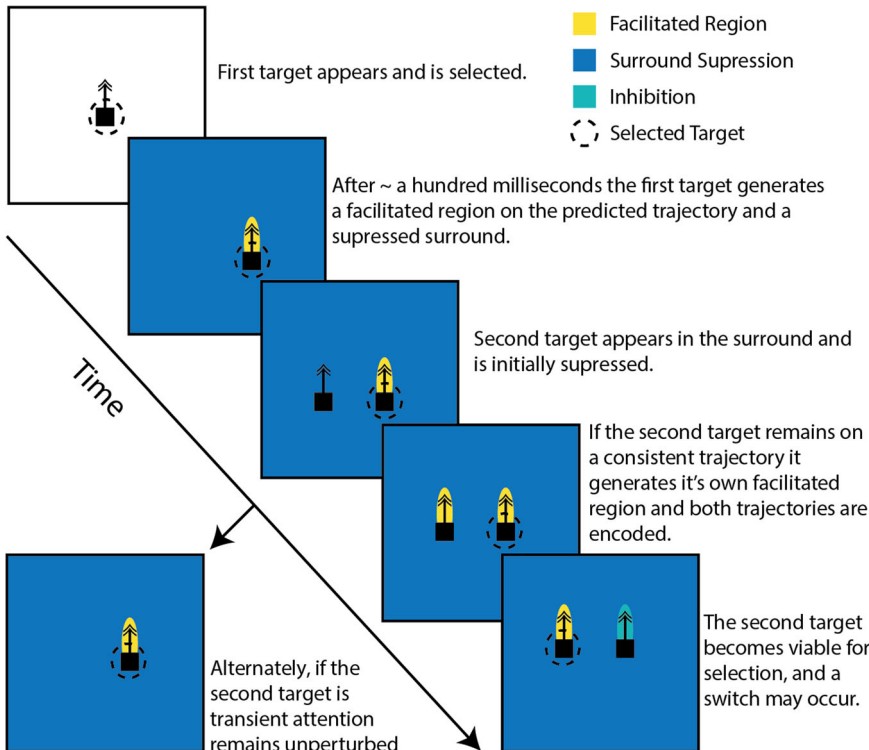

**Fig. 6 Schematic depicting the proposed interaction between Predictive Gain Modulation and target selection.** Predictive Gain Modulation may act as a gatekeeping mechanism for selection by suppressing transient stimuli that appear elsewhere within the excitatory receptive field, but passing consistent stimuli that may be of potential behavioural relevance. This ensures the attention system does not get captured by transiently highly-salient distractions, but is still afforded the flexibility to track and respond to novel stimuli.

we observed that the strength of this biasing diminishes over time and that endogenous switches in attentional selection increase in probability as both targets remain on continuous trajectories[13]. Here we have shown that non-selected targets generate facilitation and that switching results in a suppressive drive conceptually similar to Inhibition-of-Return (IOR). This allows the system to reliably switch to novel targets that appear during tracking. To our knowledge, an IOR has not been described from an invertebrate, but is an important component of computational models of winner-take-all visual attention[37] that have been extensively studied in primate behaviour and neurophysiology[35,36].

We have also shown that targets presented in CSTMD1's inhibitory receptive field elicit a facilitation-like enhancement of inhibition over time, as well as selection of either an inhibitory or excitatory target when rival targets are presented binocularly. This establishes an ocular dominance that lasts beyond the disappearance of the selected target. Examination of CSTMD1's morphology reveals a dendritic arbour corresponding to the excitatory hemifield (in the midbrain) as well as a possible input and output arborisation (on the other side of the midbrain) corresponding to the inhibitory hemifield[14], raising the possibility that CSTMD1 makes inhibitory synaptic contact (either directly, or through an interneuron) with it's contralateral equivalent. If such is the case, the observed facilitation-of-inhibition is likely driven by excitatory facilitation in the contralateral CSTMD1. However, any specifics of the neuronal architecture and behavioural functionality associated with this interplay between inhibitory and excitatory interactions in CSTMD1's response is not yet known.

We have previously speculated that CSTMD1 is involved in signalling pursuit error when a target drifts away from the visual midline[13]. Strong Inhibitory interactions between each hemispheric CSTMD1 would ensure error signals are being generated

for only one target at a time, and enable the dragonfly to quickly reorient during an active pursuit when a target crosses the visual midline from the excitatory receptive field of one CSTMD1 to the other. Critically, targets moving upwards and towards the periphery elicit the strongest facilition[17], while prior research shows targets presented in the inhibitory hemisphere elicit the strongest inhibition when moving upwards or towards the periphery[38], presumably due to facilitation-of-inhibition via the contralateral CSTMD1. These inhibitory effects are reduced when targets are directed towards the midline – i.e., when the predicted path crosses from one hemisphere to the other[17].

The generation of ocular dominance makes it impossible to address preattentive facilitation in unselected targets across the visual midline with our data, as we cannot in principle distinguish between a scenario of no facilitation at the unselected location, and a scenario where facilitation exists upstream of CSTMD1 but is masked by the long-lasting inhibition established by ocular dominance. The duration of the suppressive effect of ocular dominance is unknown, however as it lasts at least 200 ms any Probe presented during this period would be expected to self-facilitate by the time the suppression dissipates.

The described neurophysiology is consistent with dragonfly behaviour in predator-prey interactions. Dragonflies are broadly generalist predators who forage on a variety of prey[8,46,47], achieving high capture-success rates on preferred prey species[8,9], even amidst high-density swarms[48]. During target pursuits (predatory or conspecific) dragonflies can exhibit extremes of aerobatic performance[8,48–52], outcompeting the majority of their prey on basic measures of flight performance[8]. However, 3D video analysis of flight trajectories suggests some prey actively manoeuvre, and are often able to evade dragonflies several times before the predator either captures them or gives up[8,48].

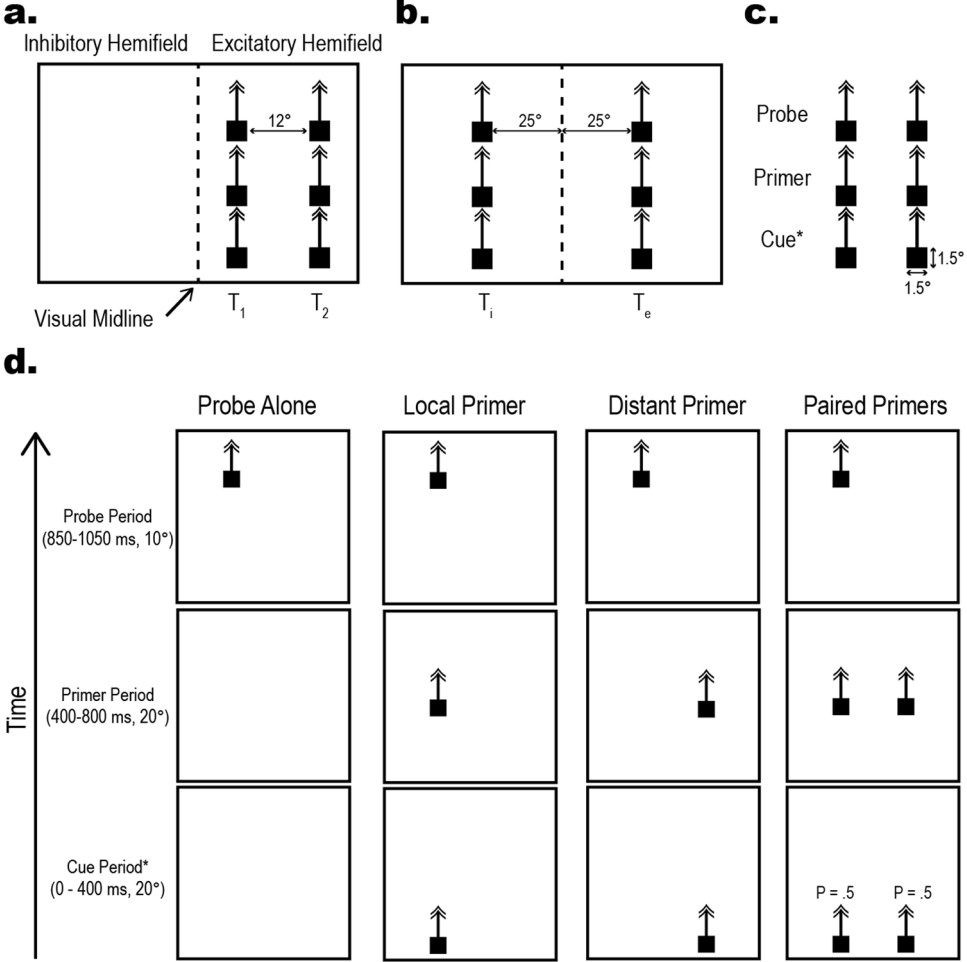

**Fig. 7 Stimulus pictograms depicting stimulus trajectories and permutations. a** Illustration (not to scale) of the target locations along two rival trajectories. Rival trajectories spaced 12° apart in the excitatory receptive field. $T_1$ refers to the target on the midline-adjacent trajectory, while $T_2$ refers to the target on the more peripheral trajectory. **b** In the cross-hemispheric experiments the trajectories are 50° apart, equidistant from the visual midline (dotted line). These targets are named $T_i$ (Target-inhibitory) and $T_e$ (Target-excitatory) based on the visual hemifield in which they reside. **c** Trajectories are divided into three vertical sections, the Cue, Primer, and Probe section. **d** Target position over time (ascending Y) in a subset of conditions representative of the base experiment. These four trials illustrate the $T_1$ trials and are intermixed with mirrored $T_2$ trials (not shown for brevity). The cross hemispheric experiments use the same basic pattern; however, trajectories are further apart and centred on the midline as described above (**b**). * The Cue Period stimuli are only presented in a subset of Cue-related experiments. In Cued, Paired-Primer trials there is an equal chance for the Cue to be present on either $T_1$ or $T_2$, but never both.

Additionally, conspecific pursuits are much longer in duration and involve rapid weaving, turns, and role reversals[52]. Pursuit success in these scenarios is lower than when targets either cannot or do not manoeuvre[8,48]. In addition to potential biomechanical limitations, reduced capture success for erratically moving targets may be related to the limits of neuronal facilitation in the STMD system. Spatial facilitation in CSTMD1 encodes targets moving on a consistent, straight trajectory and, although CSTMD1 is not direction selective, spatial facilitation establishes temporary direction selectivity[17]. Thus, prey or conspecifics moving on erratic pathways are likely to repetitively slip outside of the spotlight that attentional facilitation has generated and into the suppressed surround, impairing the neuronal response and possibly contributing to reduced capture success[48]. This so-called protean movement behaviour pattern is exhibited by many prey species[53–55] is displayed across taxa[56–59], including in insects[48,55,60–62], with some species even exhibiting protean movement pre-emptively[48,55] without knowledge of a nearby predator ('Protean Insurance'). Intriguingly, protean behaviour appears to be a successful anti-predator defence even if the

movements are technically predictable[53], such as the spiralling take-off observed in *chironomid* midges[55], so long as the prey avoids a straight trajectory that is ideal for generating neuronal facilitation. The effect of protean movement on target tracking in CSTMD1 is currently under investigation.

The dragonfly target tracking system has balanced two mutually opposing demands of attention: ignoring distractions, and flexibly responding to novel stimuli, by utilizing Predictive Gain Modulation as a gate-keeping mechanism. We have shown that when a pair of targets is presented in the excitatory receptive field, both targets generate a spotlight of facilitatory gain enhancement as they move along a linear trajectory, despite only one being attentionally selected for active representation in the spiking response of CSTMD1. This implies that target facilitation computations occur upstream of target selection in the STMD network. In addition to potential roles for ensuring robust responses in dynamically varying environments[18] and driving target trajectory predictions[17], we suggest that facilitation of non-selected targets represents a strategy for low-level, passive target tracking of potentially important stimuli without negatively

influencing encoding of the single target ultimately selected for the direction of behaviour.

## Methods

**Experiment Preparation**. We recorded from 33 wild-caught male *Hemicordulia sp.* No ethical approval is required for the use of Dragonflies under the Australian code for the care and use of animals for scientific purposes. Dragonflies were immobilized to an articulating magnetic stand with a 1:1 wax-rosin mixture. The head was tilted forward to allow access to the back of the head capsule and a small hole dissected in the chitin over the lobula complex of the left optic lobe. We pulled aluminosilicate electrodes (Harvard Apparatus) using a Sutter Instruments P-97 and filled them with a 2 M KCL solution. Electrodes were inserted into the brain with the perineuronal sheath in-tact (aside from the puncture zone) using a piezo-electric stepper for a typical resistance of 40–140 MΩ. Intracellular responses were digitized at 10 kHz for offline analysis.

CSTMD1 was identified based on well-characterised electrophysiological properties, including small target selectivity (lack of response to grating or bar stimuli), receptive field size (large field), shape (discrete excitatory and inhibitory components, separated along the midline), and lack of strong directional selectivity, as well as spike properties such as firing rate, amplitude, and waveform shape.

**Experimental design and visual stimuli**. Visual stimuli were presented on a high-definition LCD monitor (refresh rate: 165 Hz) using a custom-built presentation and data acquisition suite based on MATLAB (RRID: SCR_001622) and Psychtoolbox (RRID: SCR_002881. Available at: www.psychtoolbox.org). The animal was placed 20 cm from the monitor and centred on the visual midline in order to minimize off-axis artefacts.

Time information is given in milliseconds (ms), size is given in angular degrees (°) relative to the insect, and speed is given as °/second (°/s).

Depending on the trial condition, either single or paired targets (1.5° by 1.5° squares) were presented at 12° or 50° horizontal separation ascending the visual display at 50°/s. Descriptive pictograms of visual stimuli, trial permutations, and stimulus terminology are presented in Fig. 7. Within CSTMD1's excitatory receptive field, we define Target-1 ($T_1$) as nearer to the midline and Target-2 ($T_2$) as more peripheral (Fig. 7a, left). When presented in each visual hemifield (25° equidistant from the visual midline), we refer to Target-Inhibitory ($T_i$) and Target-Excitatory ($T_e$), with respect to the excitatory and inhibitory regions of the receptive field (Fig. 7a, right). In experiments testing facilitation, target trajectories were vertically divided into 3 sections: Probe, Primer, and Cue, leading to 6 possible trajectory segments; 3 vertical locations on 2 contiguous trajectories (Fig. 7b). For selective attention experiments not addressing facilitation, we refer more simply to a Target on a described trajectory[12,38]. To limit habituation, experimental trials were randomly interleaved with at least a 12 s rest period.

To measure the degree of facilitation, a 200 ms Probe target (Weber contrast = 1) was presented either alone (Probe Alone control conditions) or following one Primer target. For Probes presented within the excitatory receptive field, we counted spikes within a 100 ms window, 50 ms offset from the Probe onset to account for response delays.

To generate facilitation of the Probe target, Primer targets were presented for 400 ms (Fig. 7d, top row). Primer targets were either; absent for a Probe Alone, no-facilitation condition; matched to the Probe trajectory in the Local Primer condition; on a horizontally offset (unmatched) trajectory in the Distant Primer condition; or presented simultaneously at both matched and unmatched trajectories in a Paired Primers condition (Fig. 7d, middle row). On a subset of trials (50%) we introduced an attentional cue (400 ms; Fig. 7d, bottom row), which biased selection towards either the $T_1$ or $T_2$ Primer[13]. In all trials, the selection was determined by a Selectivity metric (see below), not assumed on the basis of cue locations.

**Analysis of selective attention**. To allow identification of which Primer was selected during paired trials, we frequency-tagged targets[13]. Each target's contrast was modulated (0.22 to 1) at a unique frequency (11, 15 Hz) with a square waveform, resulting in frequency-locked responses. This allowed us to read-out the selected Primer on ~70% of Paired Primer trials. On the remaining 30% of trials, frequency-tagging did not elicit sufficient modulation, likely due to neuronal habituation or saturation. These trials where selection could not be identified were excluded from further analysis. We used continuous Wavelet Transforms (Analytic Morse wavelet, gamma = 3) to extract pseudo-frequency information from the Inverse interspike interval (ISI), which represents the instantaneous spike rate of the neuronal response[13]. At frequencies corresponding to $T_1$ and $T_2$ target contrast modulation, we average wavelet output across time to yield a measure of responsiveness to each frequency ($T1_r$, $T2_r$). Selectivity was defined as:

$$Selectivity\ (S) = T1_r^2 / \sqrt{T1_r^2 + T2_r^2} - \sqrt{T1_r^2 - T2_r^2} \tag{1}$$

Yielding a single-value metric ranging between +1 (selection of $T_1$) and −1 (selection of $T_2$), where ~0 would be indicative of a switch in attention during the trial[13]. We applied a selectivity threshold of ± 0.3 to discard trials with unreliable target identification.

For a robust wavelet measure, we required at least 400 ms of continuous data. As CSTMD1 is known to switch selected targets in less than 400 ms[12,13] and that biasing effects diminish over time[13], we additionally weighted Selectivity closer to the Probe onset.

$$Weighted\ Selectivity = \sum_t S_t * 2 * t/n \tag{2}$$

Where t = time point and n = total time points.

**Statistics and reproducibility**. As selective attention operates on a trial-by-trial basis, any given trial is independent and averaging across the trials (technical replicates) would mask the observation. To ensure statistical robustness, we repeated experiments across several dragonflies (total = 33). We use n to denote the number of trials and additionally report the number of dragonflies for each experiment.

All data analysis was conducted in MATLAB R2019a (RRID: SCR_001622), including the Wavelet Toolbox. We report exact P except when <0.001. For all datasets, statistical outliers >5 * the standard deviation have been removed (3 trials total). We additionally report Hedge's g [95% confidence interval] as a measure of effect size. Unless otherwise stated all hypothesis tests use the Wilcoxon rank-sum test and are corrected for multiple comparisons (Bonferroni-holm correction). Boxplots illustrate the Median, IQR, and maximum range data points not considered statistical outliers. Additionally, all trials (including statistical outliers< 5* the standard deviation) are plotted as individual data points.

**Reporting summary**. Further information on research design is available in the Nature Research Reporting Summary linked to this article.

## Data availability

Data was stored and analysed in MATLAB. The data presented in this study are available via figshare at DOI: 10.25909/19407572.

## Code availability

MATLAB code used to analyse the data in this study is available via figshare at DOI: 10.25909/19407572.

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

## Acknowledgements

This work was supported by the Australian Research Council's Future Fellowship Scheme (FF180100466), the Australian Government Research Training Program (RTP), the Swedish Research Council (VR 2014-4904 and VR 2018-03452), and the Swedish Foundation for International Cooperation in Research and Higher Education (STINT). We thank the manager of the Adelaide Botanic Gardens for allowing insect collection. We also thank Patrick Shoemaker for valuable feedback on an early version of the manuscript.

## Author contributions

B.H.L.: Designed experiments, collected data, analysed data, wrote the manuscript. B.J.E.E.: Designed experiments, collected data, analysed data, and manuscript feedback. J.M.F.: Designed experiments, analysed data, manuscript feedback. D.C.O.: Designed experiments, analysed data, manuscript feedback. S.D.W.: Designed Experiments, analysed data, manuscript feedback.

## Competing interests

The authors declare no competing interests.
