## [Peer Review File · Communications Biology]

Reviewers' comments:

Reviewer #1 (Remarks to the Author):

Review of Lancer et al., COMMSBIO-21-2986-T, 'Preattentive Facilitation and Inhibition of Return in a Dragonfly Visual Neuron'

This is an interesting study of the target motion encoding properties of an identified visual neuron, CSTDM1, in dragonfly. The work centers on an interesting question in predatory brain function, which is attentional stability versus flexibility of target selection. This is addressed in the context of continuous vs novel moving targets within the cell's receptive field (RF). The 'priming' phenomenon that is central to the work is demonstrated clearly by the results in Figure 1, and subsequent experiments build upon it in a clear and well-motivated, step-by-step manner. Using temporally-tagged stimuli and cued primer selection paradigms, the authors demonstrate that two, paired primers within the excitatory receptive hemifield facilitate the response, even if the selected primer is discontinuous with the subsequent target probe. This demonstrates that facilitation is not tightly spatially restricted, and that near-targets do not inhibit the selected target. They report that switching target selection from a cued, local to a distal primer reduced the response to the test probe. This is interpreted as an mechanism for 'inhibition of return', a known psychophysical phenomenon, which appears reasonable, but see point 6, below. Finally, an intriguing feature of the CSTDM1 receptive field is that it comprises an excitatory and contralateral, inhibitory hemifield. Earlier work showed that continuous target motion from the inhibitory into the excitatory hemifield facilitates the response in the excitatory hemifield, indicating that processing of the facilitation involves additional, possible presynaptic circuitry, and the extent of this circuit and its role in target selection across the two hemifields remains unclear. The authors provide new information about signal integration across the hemifields by repeating the single and paired primer paradigm used within the excitatory hemifield now with cues and primers presented in the two hemifields. Different from facilitation following selection of the distant primer within the excitatory hemifield, shown in the first section of the manuscript, selection of the distant primer in the inhibitory hemifield strongly suppressed the response to the target probe. Since the excitatory and inhibitory hemifields are in opposite visual fields, this is interpreted as a form of ocular dominance to facilitate target selection stability, which is reasonable. In all, the experiments are well designed and present a concise set of assays that test receptive field function and nonlinear visual target encoding in CSTDM1. My main concern with the manuscript in its current form is that it is difficult to follow: cue, primer, probe, selected vs. non-selected, local vs distant, later excitatory vs inhibitory – there are a lot of stimulus perturbations and selection outcomes here. It would likely help the general audience to spell out the various cases in more detail. As is, Results are presented so concise that it is difficult to follow how results lead to a conclusive statement, and how that leads to the next assay. Some of this is reflected in my comments, below. I think the question is interesting and the recordings are excellent, but I struggle with the presentation. Diagrams to illustrate paradigm/hypothesized working models would be useful throughout. Now, we have to wait until Figure 6, and that model could be improved – see my point 10, below.

I have the following questions/comments.

1. How was CSTMD1 identified in recordings?
2. Results and Figure 1A: Trajectories T1 and T2 are introduced as 'rival trajectories' within CSTMD1's receptive field. Legend says the receptive field was mapped with horizontal motion; this is vertical motion. How spatially linear is the receptive field?
3. Frequency tagging of primer target is clever (this is a compliment; not a concern).
4. Results, line 107: 'Thus, we observed neuronal facilitation even on the trajectory of non-selected targets during Paired Primer conditions'. It is not clear from the results listed leading up to this statement how facilitation of the probe response followed selection of the local or distant primer. This would be based on the frequency analysis, but that is not shown in the figure, nor made explicit in the text. Please elaborate/explain. This statement appears to be supported by the paragraph that follows, and Figure 2 – check if, indeed, this statement is premature at its current

location.

5. Figure 2A: Please confirm, and make explicit in the figure legend, that this is data from a single example trial.

6. Results, Line 139: Paragraph opens with 'What could account for the weaker facilitation in this case?'. Because the preceding paragraph ends with results for several conditions/selection outcomes, it is not clear exactly what is the weaker facilitation that is referenced here. Please make explicit in this opening sentence that you next aim to explain '... the weaker facilitation in those cases where primer X was selected and response to probe Y was smaller', or something to that effect – clearly identify what 'case' you are referring to here.

7. Results, line 159: 'This reveals that facilitation generated by the introduction of a novel (i.e., not cued) target appearing during tracking of a cued target is an all-or-nothing effect'. It is not clear what this statement is based on. All or nothing implies a bimodal distribution. Figure 3B shows a broad distribution. Please explain.

8. Line 210: Inhibition will be apparent only for a non-zero spontaneous spike rate. Was there sufficient spontaneous activity to test for inhibition with a target/probe in the inhibitory hemifield?

9. Related, The CSTMD1 receptive field map shown in Figure 1 shows a lowest response rate of zero in the inhibitory hemifield. However, if there is spontaneous activity, then a stimulus in the inhibitory hemifield would suppress this, as suggested in the text, and the *response* to this stimulus should be represented with a negative rate. Otherwise, consider labeling the c-axis 'Firing rate', not Response rate – because the latter implies the *change* in spike rate from spontaneous following stimulus presentation.

10. The schematic of Figure 6 does not differentiate between the excitatory and inhibitory hemifield. If this is intentional, because the effect of cueing and target selection act the same way in each, it would still be helpful to make that explicit. This is particularly important because cross-hemifield interaction is an apparent key novel part of this study.

Reviewer #2 (Remarks to the Author):

In this manuscript, Lancer and collaborators conduct intracellular recordings of an identified wide-field, target-detecting neuron found in dragonflies.

Building on previous work, they investigate in further detail how the CSTMD1 neuron selectively responds to only one of two small, moving visual stimuli (targets), and how these responses are modulated by preceding stimuli.

Previously, it was shown that responses are enhanced by preceding target motion which moves along the same trajectory, but not for a distant trajectory, demonstrating that the enhancement (or facilitation) acts locally rather than globally.

Here, the authors report that when local and distant preceding target motion are presented on-screen together, the neuron's response to target motion is subsequently increased, regardless of which preceding target it responded to.

They then report that targets moving in the inhibitory receptive field are also subject to an analogous facilitation of inhibition, which decreases the response to targets in the opposite hemifield when they are preceded by additional motion along the same trajectory. This is congruent with the idea that facilitation originates upstream of CSTMD1 (with similar inputs to both sides of the neuron, acting in an inhibitory manner on one side, excitatory on the other) and that upstream, facilitation is a local effect.

However, a further set of experiments seems to suggest that facilitation of inhibition acts more globally than that of excitation.

The work presents ambitious experiments, which appear to have been carried out with great technical skill, using an elegant method for identifying which target the neuron selectively responded to. They further demonstrate the complex physiological responses of this cell in

demanding visual tasks.

While not a quantum leap forward, the results should be valuable for elucidating the underlying mechanisms leading to the observed responses in CSTMD1 and, more widely, for understanding visual processing supporting target pursuit in flying insects.

I find no critical issues, but have a number of quibbles related to the terms used in the title, some technical questions, and a complaint about the clarity in the presentation of later results.

1. Preattentive facilitation and inhibition of return (IOR) as properties of CSTMD1

Preattentive facilitation is not defined in the manuscript. I think this refers to the facilitation of unselected targets. As the authors themselves state (L277, L358), facilitation is likely taking place upstream, rather than being a property of CSTMD1 itself. The representation of all small-targets in all sensitive neurons downstream of the facilitation then might be expected to exhibit facilitated responses, not just in CSTMD1. Since the term is only used twice (in the title and abstract), it doesn't seem to serve a useful purpose.

The discussion of IOR (L64-65) as a potentially similar mechanism found in human psychophysics is interesting and adds color to the interpretation of CSTMD1's functional role. I do not believe that the effects reported here should be termed IOR though, for the following reasons:

- i) a comparison between two of the conditions in panels Fig. 3A and 3B presents a good test of IOR: according to classic experiments, IOR should be evident when the neuron responds to local->distant->local (A, cyan) (which the authors seem to acknowledge in L145-146), but not when it responds to distant->distant->local (B, cyan) (since in this case the local target/probe has not previously been attended). With IOR, a lower mean response in A would be expected, whereas the reported means of A and B are approximately equal.
- ii) the effect which is reported as IOR is a weaker facilitation, rather than a suppression or inhibition of the neuron. If the response upon return to a previously attended target is greater than the response to a novel target, IOR does not seem to be effective.
- iii) repeated references to IOR in the interpretation of results starts to give the reader the impression that CSTMD1 is performing a sort of visual search, like scanning faces in a crowd, while all of the results here demonstrate that the neuron tends to continue to track its currently selected target and not switch unless it disappears. If differing visual targets were used, perhaps a greater degree of switching would occur, until settling on an optimal target.

2. There are many trials visible in Fig. 1-3 where the probe response is exactly 0 sp/s, including 'probe alone trials'. They obviously greatly affect the reported sample mean in some cases. Can the authors explain these? Do they correlate with the trial number, as an effect of habituation? Should they not be excluded?

3. Also, how does recording the same neuron in an individual dragonfly multiple times affect target switching? I am not entirely convinced that multiple trials from an individual (a statistical unit) can be treated as independent measures (L420-421). Was a method for treating this considered, such as linear mixed models or hierarchical bootstrap (see Saravanan et al., 2020)?

4. Only 100 ms of response is analyzed for a 200 ms probe presentation, during the period where the response is still increasing. Can the authors justify this? Do the findings change if the steady-state firing rate is used? Also, was it considered to examine the rate of increase under different conditions (which would be more akin to the qualities measured which show IOR in humans)?

5. I found reading the text relating to Fig. 4-5 long and very hard work. I would suggest focussing on one or two of the most important findings here, and presenting them in a straightforward manner. Some of the extra details (5A, for example) could be moved to supplementary material.

6. Throughout, many terms are used to describe a target moving on-screen at a particular position or time. The stimulus schematics could be much more helpful if the different regions (cue/primer/probe, local/distant) were labeled. The two parallel arrows in Fig. 3 and 5 are also initially confusing (Fig. 4D's overlapping arrows are better), and the color-coded dashed circles suggest that these are the regions analyzed, rather than the probe region.

7. In Fig. 5 the responses to targets in the inhibitory receptive field seem to be lower than 10 spikes/s, i.e. around 1 or 2 spikes in response to a 100 or 200 ms primer. Could the authors comment on how frequency tagged targets can still be identified in this condition? Or is low vs high firing rate the distinguishing property?

Minor points:

L24: why and how IOR should be an important attribute for target pursuit is not further discussed

L32: 'Selective attention' as important to many species across taxa would require a reference

L137-138: "[...] indicating strong facilitation in at least a subset of trials." It is not incorrect but it sounds a bit biased since only two points lie below the mean (gray dashed line)

L163: "Such a an"

L171: "((Figure 2))"

L192: "d = 1.11" Hedge's g?

Caption of Fig. 3 explains what the gray dashed line crossing the box plots means (i.e. probe alone mean) but it is already present in Fig. 1C,D without explanation

Reviewer comments presented in 'normal text.'

Author responses presented in 'blue text.'

Referee expertise:

Referee #1: neuron, retinal diseases, cellular and molecular neuroscience

Referee #2: fly visual system, neurons

Reviewers' comments:

We thank the reviewers for their detailed comments and time. Our responses are included on a point-by-point basis throughout.

Reviewer #1 (Remarks to the Author):

Review of Lancer et al., COMMSBIO-21-2986-T, 'Preattentive Facilitation and Inhibition of Return in a Dragonfly Visual Neuron'

This is an interesting study of the target motion encoding properties of an identified visual neuron, CSTDM1, in dragonfly. The work centers on an interesting question in predatory brain function, which is attentional stability versus flexibility of target selection. This is addressed in the context of continuous vs novel moving targets within the cell's receptive field (RF). The 'priming' phenomenon that is central to the work is demonstrated clearly by the results in Figure 1, and subsequent experiments build upon it in a clear and well-motivated, step-by-step manner.

Using temporally-tagged stimuli and cued primer selection paradigms, the authors demonstrate that two, paired primers within the excitatory receptive hemifield facilitate the response, even if the selected primer is discontinuous with the subsequent target probe. This demonstrates that facilitation is not tightly spatially restricted, and that near-targets do not inhibit the selected target. They report that switching target selection from a cued, local to a distal primer reduced the response to the test probe. This is interpreted as an mechanism for 'inhibition of return', a known psychophysical phenomenon, which appears reasonable, but see point 6, below.

Finally, an intriguing feature of the CSTDM1 receptive field is that it comprises an excitatory and contralateral, inhibitory hemifield. Earlier work showed that continuous target motion from the inhibitory into the excitatory hemifield facilitates the response in the excitatory hemifield, indicating that processing of the facilitation involves additional, possible presynaptic circuitry, and the extent of this circuit and its role in target selection across the two hemifields remains unclear. The authors provide new information about signal integration across the hemifields by repeating the single and paired primer

paradigm used within the excitatory hemifield now with cues and primers presented in the two hemifields. Different from facilitation following selection of the distant primer within the excitatory hemifield, shown in the first section of the manuscript, selection of the distant primer in the inhibitory hemifield strongly suppressed the response to the target probe.

Since the excitatory and inhibitory hemifields are in opposite visual fields, this is interpreted as a form of ocular dominance to facilitate target selection stability, which is reasonable.

In all, the experiments are well designed and present a concise set of assays that test receptive field function and nonlinear visual target encoding in CSTMD1. My main concern with the manuscript in its current form is that it is difficult to follow: cue, primer, probe, selected vs. non-selected, local vs distant, later excitatory vs inhibitory – there are a lot of stimulus perturbations and selection outcomes here. It would likely help the general audience to spell out the various cases in more detail. As is, Results are presented so concise that it is difficult to follow how results lead to a conclusive statement, and how that leads to the next assay. Some of this is reflected in my comments, below. I think the question is interesting and the recordings are excellent, but I struggle with the presentation. Diagrams to illustrate paradigm/hypothesized working models would be useful throughout. Now, we have to wait until Figure 6, and that model could be improved – see my point 10, below.

On the point of clarity, we have included a description & associated figure of stimulus components in order to both enhance first-pass clarity and serve as a reference throughout the manuscript Line.

I have the following questions/comments.

1. How was CSTMD1 identified in recordings?

CSTMD1 was identified on the basis of well-known physiological response properties, such as receptive field shape and size, spike properties, and spike rate. We have expanded on the identification process in the methods section of the manuscript. (Line 399-401).

2. Results and Figure 1A: Trajectories T1 and T2 are introduced as 'rival trajectories' within CSTMD1's receptive field. Legend says the receptive field was mapped with horizontal motion; this is vertical motion. How spatially linear is the receptive field?

CSTMD1 has a homogenous receptive field that responds to targets of appropriate size and speed in all directions, with slight variation in firing rate that preferences targets moving up and to the right. For interest, comparable vertical Receptive fields can be found in Bolzon (2009), as cited within this manuscript.

3. Frequency tagging of primer target is clever (this is a compliment; not a concern).

Thank you! We cannot take full credit; frequency-tagging was inspired by a similar technique used in Electroencephalogram and Local Field Potential research, where it is known as the Steady-State Visual Evoked Potential.

4. Results, line 107: 'Thus, we observed neuronal facilitation even on the trajectory of non-selected targets during Paired Primer conditions'. It is not clear from the results listed leading up to this statement how facilitation of the probe response followed selection of the local or distant primer. This would be based on the frequency analysis, but that is not shown in the figure, nor made explicit in the text. Please elaborate/explain. This statement appears to be supported by the paragraph that follows, and Figure 2 – check if, indeed, this statement is premature at its current location.

We have re-worded and expanded this sentence to enhance clarity and better reflect accurate conclusions. We have also re-titled Figure 1 and Figure 2 to reflect the flow of discovery better. (Sentence now begins line 110).

5. Figure 2A: Please confirm, and make explicit in the figure legend, that this is data from a single example trial.

We have updated the figure legend.

6. Results, Line 139: Paragraph opens with 'What could account for the weaker facilitation in this case?'. Because the preceding paragraph ends with results for several conditions/selection outcomes, it is not clear exactly what is the weaker facilitation that is referenced here. Please make explicit in this opening sentence that you next aim to explain '... the weaker facilitation in those cases where primer X was selected and response to probe Y was smaller', or something to that effect – clearly identify what 'case' you are referring to here.

We have edited this paragraph opening to explicitly reference the result under discussion. (Now line 143)

7. Results, line 159: 'This reveals that facilitation generated by the introduction of a novel (i.e., not cued) target appearing during tracking of a cued target is an all-or-nothing effect'. It is not clear what this statement is based on. All or nothing implies a bimodal distribution. Figure 3B shows a broad distribution. Please explain.

The Paired Primer distribution in Figure 3B (purple) is indeed broad, but matches closely the 'merged model' (purple, dashed) which is a summed combination of the 'Local' and 'Distant' Primer conditions. This suggests that the trials comprising the Paired Primer

distribution are approximately evenly drawn from a facilitated distribution (matching the Local Primer), or an unfacilitated distribution (matching the Distant Primer). As both underlying distributions have reasonable overlap, the resulting merged distribution is quite broad (See the Merged Model), which matches the empirical data (purple line).

To avoid this confusion, we have replaced the term all-or-nothing with 'binary' to emphasise that the outcome of any one trial in this condition appears to be drawn from one of two overlapping distributions (I.e., Local or Distant Primer)

8. Line 210: Inhibition will be apparent only for a non-zero spontaneous spike rate. Was there sufficient spontaneous activity to test for inhibition with a target/probe in the inhibitory hemifield?

We found spontaneous activity to be around 5-10 sp/s. With an inhibitory target, this spontaneous rate is regularly driven to 0 sp/s as observed in Fig 4F. We have added a sentence describing this.

9. Related, The CSTMD1 receptive field map shown in Figure 1 shows a lowest response rate of zero in the inhibitory hemifield. However, if there is spontaneous activity, then a stimulus in the inhibitory hemifield would suppress this, as suggested in the text, and the *response* to this stimulus should be represented with a negative rate. Otherwise, consider labeling the c-axis 'Firing rate', not Response rate – because the latter implies the *change* in spike rate from spontaneous following stimulus presentation.

This is correct – we have updated the figure axis to read firing rate.

10. The schematic of Figure 6 does not differentiate between the excitatory and inhibitory hemifield. If this is intentional, because the effect of cueing and target selection act the same way in each, it would still be helpful to make that explicit. This is particularly important because cross-hemifield interaction is an apparent key novel part of this study.

This schematic refers particularly to interactions within the excitatory receptive field, as the novel finding of ocular dominance when targets are presented across hemispheres masks our ability to answer the original question for this condition. We have made this point more explicit in the figure caption, as well as moved early discussion points relating to the inhibitory hemifield experiments later in the discussion and expanding on this point with a new paragraph.

Reviewer #2 (Remarks to the Author):

In this manuscript, Lancer and collaborators conduct intracellular recordings of an identified wide-field, target-detecting neuron found in dragonflies.

Building on previous work, they investigate in further detail how the CSTMD1 neuron selectively responds to only one of two small, moving visual stimuli (targets), and how these responses are modulated by preceding stimuli.

Previously, it was shown that responses are enhanced by preceding target motion which moves along the same trajectory, but not for a distant trajectory, demonstrating that the enhancement (or facilitation) acts locally rather than globally.

Here, the authors report that when local and distant preceding target motion are presented on-screen together, the neuron's response to target motion is subsequently increased, regardless of which preceding target it responded to.

They then report that targets moving in the inhibitory receptive field are also subject to an analogous facilitation of inhibition, which decreases the response to targets in the opposite hemifield when they are preceded by additional motion along the same trajectory. This is congruent with the idea that facilitation originates upstream of CSTMD1 (with similar inputs to both sides of the neuron, acting in an inhibitory manner on one side, excitatory on the other) and that upstream, facilitation is a local effect. However, a further set of experiments seems to suggest that facilitation of inhibition acts more globally than that of excitation.

The work presents ambitious experiments, which appear to have been carried out with great technical skill, using an elegant method for identifying which target the neuron selectively responded to. They further demonstrate the complex physiological responses of this cell in demanding visual tasks.

While not a quantum leap forward, the results should be valuable for elucidating the underlying mechanisms leading to the observed responses in CSTMD1 and, more widely, for understanding visual processing supporting target pursuit in flying insects.

I find no critical issues, but have a number of quibbles related to the terms used in the title, some technical questions, and a complaint about the clarity in the presentation of later results.

1. Preattentive facilitation and inhibition of return (IOR) as properties of CSTMD1

Preattentive facilitation is not defined in the manuscript. I think this refers to the facilitation of un-selected targets. As the authors themselves state (L277, L358), facilitation is likely taking place upstream, rather than being a property of CSTMD1 itself. The representation of all small-targets in all sensitive neurons downstream of the facilitation then might be expected to exhibit facilitated responses, not just in CSTMD1.

Since the term is only used twice (in the title and abstract), it doesn't seem to serve a useful purpose.

We have added a definition & explanation of the preattentive facilitation term to the first paragraph of the discussion section.

The discussion of IOR (L64-65) as a potentially similar mechanism found in human psychophysics is interesting and adds color to the interpretation of CSTMD1's functional role. I do not believe that the effects reported here should be termed IOR though, for the following reasons:

i) a comparison between two of the conditions in panels Fig. 3A and 3B presents a good test of IOR: according to classic experiments, IOR should be evident when the neuron responds to local->distant->local (A, cyan) (which the authors seem to acknowledge in L145-146), but not when it responds to distant->distant->local (B, cyan) (since in this case the local target/probe has not previously been attended).

We believe this is just the case: The broad cyan distribution in B matches closely with the 'merged model' distribution (purple dashed line), suggesting the cyan distribution is comprised of a sum of trials which match either the Local or Distant primer conditions (I.e. some where facilitated, and some not). The breadth of the two constituent distributions leads to very broad distribution when summed.

In contrast the cyan distribution in A shows a narrower breadth that matches an intermediate response rather the merged model.

However, we agree with the reviewer's overall account & criticisms below (particularly ii), thus we have edited the manuscript to avoid strong declarations of Inhibition of Return (including in the title), whilst keeping discussion points relating to IoR as an interesting comparison.

With IOR, a lower mean response in A would be expected, whereas the reported means of A and B are approximately equal.

ii) the effect which is reported as IOR is a weaker facilitation, rather than a suppression or inhibition of the neuron. If the response upon return to a previously attended target is greater than the response to a novel target, IOR does not seem to be effective.

iii) repeated references to IOR in the interpretation of results starts to give the reader the impression that CSTMD1 is performing a sort of visual search, like scanning faces in a crowd, while all of the results here demonstrate that the neuron tends to continue to track its currently selected target and not switch unless it disappears. If differing visual

targets were used, perhaps a greater degree of switching would occur, until settling on an optimal target.

We have clarified this in our discussion section. The specific stimuli that elicit a switch and functional role of switching behaviour are outside of the scope of the current manuscript but provide the basis for future work.

2. There are many trials visible in Fig. 1-3 where the probe response is exactly 0 sp/s, including 'probe alone trials'. They obviously greatly affect the reported sample mean in some cases. Can the authors explain these? Do they correlate with the trial number, as an effect of habituation? Should they not be excluded?

We excluded any trials with a low (<25 sp/s) spikerate during the Primer period in order to ensure that all trials in Primer conditions were responsive and tested our hypotheses related to Primer response, however the reported spike rates are calculated on the basis of the *Probe* response, so trials which responded to a Primer but then did not respond to the Probe are still included.

We elected to not exclude trials on the basis of their Probe responses (including Probe alone trials) in order to avoid biases, for example, if any conditions involved suppression of Probe responses then these trials would be at higher risk for being removed.

Although CSTMD1 does habituate, we included at least a 15 second rest period between each trial to limit the effects of habituation. We did not find that reduced Probe response was related to trial number.

It is also worth noting that CSTMD1 sometimes stochastically exhibits 'delayed' reactions lasting a few hundred milliseconds (E.g., see traces in Wiederman & O'Carroll 2013). Since the Probe was only short (200 ms), and further we focussed analysis on a 100 ms window, some trials in which CSTMD1 exhibited a delayed response to the probe will show low or even 0 response over the narrow analysis window. Indeed, this is an important part of the system, as facilitation limits these delays and helps ensure a strong response at the time of stimulus onset.

3. Also, how does recording the same neuron in an individual dragonfly multiple times affect target switching? I am not entirely convinced that multiple trials from an individual (a statistical unit) can be treated as independent measures (L420-421). Was a method for treating this considered, such as linear mixed models or hierarchical bootstrap (see Saravanan et al., 2020)?

As selective attention occurs on a trial-by-trial basis, each trial must be treated individually. The same cell within the same dragonfly can behave differently across trials, e.g., by responding to one target in one trial, and the other target in the next

(Wiederman O'Carroll, 2013; Lancer et al., 2019). This is also evident in Figure 4C of the current manuscript, where trials (dots) within the same dragonfly (columns) and thus the same cell could give divergent responses based on cue. While it is probable selection and switching have some history dependence, the specific drivers that 'decide' which target of a pair is selected on any given trial or what factors influence switching behaviour was beyond the scope of the current manuscript (which was focussed on the generation of facilitation), but is currently under investigation.

4. Only 100 ms of response is analyzed for a 200 ms probe presentation, during the period where the response is still increasing. Can the authors justify this? Do the findings change if the steady-state firing rate is used? Also, was it considered to examine the rate of increase under different conditions (which would be more akin to the qualities measured which show IOR in humans)?

We analyse such a specific, short period for two important reasons:

- 1) Targets are able to generate self-facilitation within a short time period of a few hundred milliseconds, so analysing too-late a response period is guaranteed to find facilitation, as the probe target will have facilitated itself (as in Wiederman et al., 2017 & Fabian et al 2019).
- 2) Conversely, neural delays in stimulus onset and offset response mean analysing too-early a response period is likely to include responses to the Primer target in Primer trials (but not in Probe Alone trials, with no primer) which could easily give the illusion of facilitation.

In order to overcome these potential pitfalls we must analyse a short response period in the middle of the stimulus presentation, in line with prior research on facilitation in CSTMD1.

5. I found reading the text relating to Fig. 4-5 long and very hard work. I would suggest focussing on one or two of the most important findings here, and presenting them in a straightforward manner. Some of the extra details (5A, for example) could be moved to supplementary material.

We have gone over the text relating to these figures in order to enhance clarity. However, as Figure 5A provides the rationale for the analysis conducted in 5B & C, we were unable to remove it from the main narrative without adversely impacting the flow and readability.

6. Throughout, many terms are used to describe a target moving on-screen at a particular position or time. The stimulus schematics could be much more helpful if the different regions (cue/primer/probe, local/distant) were labeled. The two parallel arrows

in Fig. 3 and 5 are also initially confusing (Fig. 4D's overlapping arrows are better), and the color-coded dashed circles suggest that these are the regions analyzed, rather than the probe region.

This point was also raised by Reviewer 1, so we have included a new Supplementary figure to act as a Reference throughout the paper. In addition, we have brought figures 3 and 5 in line with the presentation in 4.

7. In Fig. 5 the responses to targets in the inhibitory receptive field seem to be lower than 10 spikes/s, i.e. around 1 or 2 spikes in response to a 100 or 200 ms primer. Could the authors comment on how frequency tagged targets can still be identified in this condition? Or is low vs high firing rate the distinguishing property?

Spikerate was indeed used as the distinguishing property for this experiment, rather than frequency tagging. We have added a sentence to clarify this to the text describing the experiment, as well as the figure 5 legend.

Minor points:

L24: why and how IOR should be an important attribute for target pursuit is not further discussed

We have elaborated on this in the discussion section.

L32: 'Selective attention' as important to many species across taxa would require a reference

We have added some references for this sentence.

L137-138: "[...] indicating strong facilitation in at least a subset of trials." It is not incorrect but it sounds a bit biased since only two points lie below the mean (gray dashed line)

L163: "Such a an"

L171: "((Figure 2))"

L192: "d = 1.11" Hedge's g?

Caption of Fig. 3 explains what the gray dashed line crossing the box plots means (i.e. probe alone mean) but it is already present in Fig. 1C,D without explanation

We apologize for missing these typos and thank the reviewer for pointing them out – they have been corrected.

REVIEWERS' COMMENTS:

Reviewer #1 (Remarks to the Author):

I appreciate the authors' responses to my comments and recommendations. The revisions are solid, and I have no further concerns. I believe this will be an interesting contribution toward understanding visual processing underlying dragonfly prey capture.

Reviewer #2 (Remarks to the Author):

The authors have adequately addressed my concerns.

Reviewer #3 (Remarks to the Author):

Thank you for making these changes.